# Assimilating FY-4A Lightning and Radar Data for Improving Short-Term Forecasts of a High-Impact Convective Event with a Dual-Resolution Hybrid 3DEnVAR Method

Peng Liu [1], Yi Yang [1,*], Anwei Lai [2], Yunheng Wang [3,4], Alexandre O. Fierro [3,4], Jidong Gao [3] and Chenghai Wang [1]

1   Key Laboratory of Climate Resource Development and Disaster Prevention in Gansu Province, College of Atmospheric Sciences, Lanzhou University, Lanzhou 730000, China; liup16@lzu.edu.cn (P.L.); wch@lzu.edu.cn (C.W.)
2   Hubei Key Laboratory for Heavy Rain Monitoring and Warning Research, Institute of Heavy Rain, China Meteorological Administration, Wuhan 430205, China; laianwei@whihr.com.cn
3   NOAA/National Severe Storms Laboratory, Norman, OK 73072, USA; yunheng.wang@noaa.gov (Y.W.); alex.fierro@noaa.gov (A.O.F.); jidong.gao@noaa.gov (J.G.)
4   Cooperate Institute for Mesoscale Meteorological Studies, University of Oklahoma, Norman, OK 73072, USA
*   Correspondence: yangyi@lzu.edu.cn

**Abstract:** A dual-resolution, hybrid, three-dimensional ensemble-variational (3DEnVAR) data assimilation method combining static and ensemble background error covariances is used to assimilate radar data, and pseudo-water vapor observations to improve short-term severe weather forecasts with the Weather Research and Forecast (WRF) model. The higher-resolution deterministic forecast and the lower-resolution ensemble members have 3 and 9 km horizontal resolution, respectively. The water vapor pseudo-observations are derived from the combined use of total lightning data and cloud top height from the Fengyun-4A(FY-4A) geostationary satellite. First, a set of single-analysis experiments are conducted to provide a preliminary performance evaluation of the effectiveness of the hybrid method for assimilating multisource observations; second, a set of cycling analysis experiments are used to evaluate the forecast performance in convective-scale high-frequency analysis; finally, different hybrid coefficients are tested in both the single and cycling experiments. The single-analysis results show that the combined assimilation of radar data and water vapor pseudo-observations derived from the lightning data is able to generate reasonable vertical velocity, water vapor and hydrometeor adjustments, which help to trigger convection earlier in the forecast/analysis and reduce the spin-up time. The dual-resolution hybrid 3DEnVAR method is able to adjust the wind fields and hydrometeor variables with the assimilation of lightning data, which helps maintain the triggered convection longer and partially suppress spurious cells in the forecast compared with the three-dimensional variational (3DVAR) method. A cycling analysis that introduced a large number of observations with more frequent small adjustments is able to better resolve the observed convective events than a single-analysis approach. Different hybrid coefficients can affect the forecast results, either in the single deterministic or cycling analysis experiments. Overall, we found that a static coefficient of 0.4 and an ensemble coefficient of 0.6 yields the best forecast skill for this event.

**Keywords:** data assimilation; lightning and radar data; dual-resolution hybrid 3DEnVAR; convective forecast

## 1. Introduction

The accuracy and timeliness of severe weather forecasts are critical to safeguard life and property. Numerical weather prediction (NWP) models still face many challenges in accurately forecasting high-impact weather events such as the existence of biases and errors contained in the initial conditions, which are often derived or downscaled from

larger-scale model data [1,2]. Reducing the initial condition biases and errors through data assimilation is thus crucial to improve forecast skill [3–5].

Lightning data from ground-based networks and spaceborne optical instruments are able to identify areas of deep, mixed-phase convection [6–10]. The most tornadic storms (80% or more) have an increase in total flash rates near the time of the tornado, and the increase in total flash rates is often dominated by intracloud flashes [6]. The comparison analysis of lightning data and radar echoes suggests that lightning data can be used to determine the convective activity and its development probability and intensity [7]. The Geostationary Lightning Mapper (GLM) and Lightning Mapping Imager (LMI) mounted on the Geostationary Operational Environmental Satellite R-series (GOES-R) and Fengyun-4A (FY-4A) are useful for early predictions of storms and severe convection events, respectively [9,10]. Lightning has been employed to estimate convective precipitation as well [11–13]. Relying on the extensive lightning observation network and on the robust association between lightning and deep convection, the use of lightning data assimilation (LDA) is very useful to improve severe convective weather forecasting. The basic concept of LDA is to adjust selected model state variables known to be associated or well-correlated with lightning.

In most LDA research, the model state variables, such as thermal, dynamic and mass fields are usually adjusted. In the thermal field adjustment schemes, the model's latent heating profiles are nudged to rainfall rates derived from lightning observations combined with Special Sensor Microwave/Imager (SSM/I) rain-rate field and infrared (IR) brightness temperatures [14]. On this basis, a more realistic relationship between the convective rainfall rate and lightning rate was considered [15] and was implemented in the Kain-Fritsch (KF) convective parameterization scheme (CPS) [16]. The low levels of the atmosphere where lightning occurs are warmed based on parcel theory [17].

In the dynamic field adjustment schemes, based on two functional relationships between the frequency of lightning and cloud top height, cloud top height and maximum updraft, the maximum vertical velocity-driven lightning data are assimilated into the model through a nudging technique or the ensemble square root filter (EnSRF) method [18–21]. This method spreads the spatial distribution of positive vertical velocities and thus enhances the spatial distribution of severe rainfall [20]. Although it is possible to establish observation operators between temperature or vertical velocity and lightning through empirical relationships, such relationships are limited by local climate characteristics and differences in convective processes. Therefore, it is very difficult to directly adjust thermal and dynamic fields through lightning data.

In the mass field adjustment schemes, there are three main types of adjustment variables for lightning data. The first type uses water vapor as the adjustment variable. Papadopoulos et al. and Mansell et al. used lightning data to adjust the relative humidity profile or water vapor content in the cumulus parameterization scheme to activate convection [22–24]. Subsequently, a similar approach was implemented in different model [25,26]. Fierro et al. increased water vapor mass at the observed lightning locations through a continuous nudging function [27–30] or created pseudo-observations for water vapor mass based on flash-density metrics derived from lightning data which are then assimilated within a three-dimensional variational (3DVAR) system [31,32]. These adjustment schemes effectively improve the model forecast skill for convective precipitation [33–36]. The second type of mass field adjustment is based on in-cloud charging physics, which seeks to establish a relationship between ice-phase particles and flash density, and then assimilate these via nudging or ensemble assimilation methods [37–43]. Finally, the third type is based on the empirical relationship between lightning and radar reflectivity. The flash density is converted into three-dimensional radar reflectivity fields, which are then assimilated into the model [44–47].

When assimilating lightning data through proxy radar reflectivity, there are inherent errors associated with this conversion, which are dependent on the performance of the radar data assimilation (RDA) system. Observational and laboratory studies have

long shown an unambiguous association between ice-phase particles/graupel content and lightning [48–50]. However, ice particles or graupel have less impact on the background water vapor environment. The innovation of ice particles and graupel does not persist long enough and generally induces downdrafts instead of sustaining updrafts. Current studies have shown that LDA through water vapor can help trigger the observed convection earlier and also better maintain longer-lived convective systems [27–32,35]. The assimilation scheme based on water vapor, however, also features important shortcomings. First, lightning data can only determine the location of deep, mixed-phase convection. Therefore, the three-dimensional pseudo-observation of water vapor derived from lightning is primarily confined in the vertical. Second, when convection is activated by the local increases in water vapor mass in the background field, if the local increase remains small, longer spin-up time is required. The converse occurs for large water vapor mass adjustments in addition to the exacerbation of any existing wet biases. Third, the water vapor increment in the analysis field must require suitable simultaneous adjustments in the thermal and dynamic fields.

Liu et al. [35] proposed to use the cloud top height as the upper limit for the adjustment of water vapor while Fierro et al. [32] confine the adjustments from the lifted condensation level up to 3 km above it. In this study, we attempt addressing the second and third limitations by using the dual-resolution hybrid three-dimensional ensemble-variational data assimilation (3DEnVAR) method [51–53] with multivariant flow-dependent background error covariances combined with radar data in the lightning assimilation procedure.

Many studies have underscored the benefits of the assimilation of radar data for shorter term forecast improvements at the cloud-scale [54–60]. Gao et al. proposed a method of dual-Doppler radar analysis based on a variational approach. Based on radar information, the circulation inside and around the storms is well analyzed [54]. Sun et al. applied the four-dimensional variational data assimilation (4DVAR) technique to a cloud-scale model and demonstrated that the variational analysis system is able to retrieve the detailed structure of wind, thermodynamics, and microphysics using either dual-Doppler or single-Doppler information [55]. In addition, the method based on ensemble Kalman Filter assimilation of radar data is developed [56,57]. In a variational framework, Lai et al. assimilated pseudo-water-vapor and potential-temperature-driven radar data to improve the precipitation forecast [58,59]. Radar radial velocity observations contain information about the horizontal wind field component, whereas radar reflectivity and dual-polarization observations provide information about the distribution of various kinds of hydrometeors. The radial velocity and hydrometeor information provided by radar data can partially offset the lack of dynamic and hydrometeors information in the water-vapor-based LDA and reduce the spin-up time. Additionally, the assimilation of radar data (i.e., zero reflectivity) can help reduce spurious hydrometeor information in the background field. The assimilation of radar data into convective-scale NWPs also has its own limitations because of the lack of water vapor information. The pseudo-observation of water vapor derived from lightning data is useful to solve the limitations. With the ongoing expansions of satellite observation networks, the assimilation of lightning data has gradually played a more important role in NWP. Therefore, the performance of combined radar and lightning data assimilation is also examined here for its impact on strong convection forecasts.

For convective-scale data assimilation, many studies have demonstrated the notable benefits of the use of flow-dependent background error covariance statistics over stationary and isotropic background error covariances. The more advanced 4DVAR with the flow-dependent background error covariance through the forward and backward models cannot be easily applied to the convective-scale problems due to its high computational cost and nonlinearity of microphysics process. The ensemble Kalman filter method can fully derive flow-dependent background error covariances from ensemble forecasts. Often, however, the ensemble sampling size is much smaller than the actual degrees of freedom needed to resolve the covariance statistics at the cloud-scale. To mitigate these drawbacks (e.g., computational cost) and simultaneously benefit from flow-dependent information, Gao et al.

developed a hybrid 3DEnVAR system with multivariate flow-dependent background error covariance [51,52]. The algorithm uses the extended control variable approach to combine the static and ensemble-derived flow-dependent background error covariance to form a hybrid covariance. This hybrid covariance not only affects the assimilation variable but also communicates the observation information to other variables.

Based on the above considerations, it is possible to alleviate some of the shortcomings in the LDA by combining radar observation information and using more advanced assimilation methods with flow-dependent background error covariances. A high-impact severe convective case influenced by the Meiyu front in the middle and lower reaches of the Yangtze River is used to examine the impact of the assimilation of spaceborne total lightning data (cloud-to-ground plus intracloud flashes) and radar data with a 3DEnVAR system. To provide a clearer assessment of the effects of different datasets and assimilation methods in the analysis field, a set of single-analysis experiments was performed first. The ability of the combined assimilation of radar and lightning data in high-frequency cycling assimilation was also evaluated by a set of cycling analysis experiments.

In Section 2, data and assimilation methods are briefly described. In Section 3, the experimental design and model description are introduced. The results and summary discussion are presented in Sections 4 and 5, respectively.

## 2. Data and Methods

### 2.1. Lightning Data

In this work, LMI data from the FY-4A geostationary satellite were used for the LDA. The LMI can continuously monitor total lightning in 1-min intervals and with a grid spacing of 7.8 km at nadir. The LMI provides three product levels, including the event, group, and flash products. In this study, the LMI event product is used because it is the basic output unit for lightning detection and can better depict the spatial propagation of lightning flashes and, hence, electrified storm regions [10,61]. A quality control procedure is used to remove events with non-good quality flag and isolated events with no adjacent detected lightning pixels. The same procedure as Liu et al. [35] was used for creating pseudo-observations of water vapor. In single-analysis experiments, the lightning events frequency was first accumulated over a 1 h period, centered on the analysis time. In the cycling analysis experiments, the lightning events frequency was accumulated over 15 min before the analysis time. When the observed 15-min lightning events frequency per grid cell exceeded zero, the relative humidity from the background in the column associated with the grid cell was adjusted to 90% only if the background value did not already exceed 90%. In other words, if the relative humidity was already greater than or equal to 90% in that column, no adjustments were made. The upper limit of the adjustment of relative humidity was determined using the cloud top height from FY-4A. The calculated lifting condensation level from the background field was approximated as the bottom limit of the adjustment similar to Fierro et al. [31]. The relative humidity of all grid points formed a three-dimensional pseudo-observation field that was used to assimilate lightning information. Because relative humidity is proportional to the ratio between water vapor and the saturation water vapor, pseudo-observation of water vapor of lightning data can be derived equivalently via relative humidity.

### 2.2. Radar Data

A total of 37 radars were used in this study. These radars cover all the main storms during the target analysis period. The frequency of radar data is approximated to 6 min, and the radar data closest to the analysis time are assimilated. Prior to assimilation, the radar data were quality controlled; this procedure includes removing radar clutter and non-meteorological reflectivity, the removal of isolated points, and de-aliasing radial velocity [62,63]. After quality control, the radar data were interpolated to the model grid. For reflectivity, the largest value was used in the grid points where data from multiple radars overlap.

### 2.3. Data Assimilation Methods

In this study, the 3DVAR and dual-resolution hybrid 3DEnVAR methods were used to assimilate lightning and radar data (radial velocity and reflectivity factor). Both assimilation methods have shown promise at convective scales for the assimilation of lightning and/or radar data [64–68]. The two assimilation methods are briefly described in the following section.

### 2.3.1. DVAR Method

Details of the 3DVAR scheme used in this work follow Gao et al. [66,67]. The cost function is defined as follows:

$$J(\mathbf{x}) = \frac{1}{2}\left(\mathbf{x} - \mathbf{x}^b\right)^T \mathbf{B}^{-1}\left(\mathbf{x} - \mathbf{x}^b\right) + \frac{1}{2}[H(\mathbf{x}) - \mathbf{y}^o]^T \mathbf{R}^{-1}[H(\mathbf{x}) - \mathbf{y}^o] + J_c(\mathbf{x}), \qquad (1)$$

where $\mathbf{x}$ is the state vector, and $\mathbf{x}^b$ is the background state vectors, respectively; $\mathbf{y}^o$ is the observation vector; $\mathbf{B}$ and $\mathbf{R}$ are the background and observation error covariance metrics, respectively; $H$ is the observation operator; and the term $J_c(\mathbf{x})$ is any penalty or dynamic equation constraint term. The divergence equation constraint was used here. Following Gao et al. [51], an alternative control variable $\mathbf{v}$, in which $\Delta\mathbf{x} = \mathbf{B}^{\frac{1}{2}}\mathbf{v} = \left(\mathbf{x} - \mathbf{x}^b\right)$, was defined. Through this variable transform, the cost function was converted into the following preconditioned incremental form:

$$J(\mathbf{v}) = \frac{1}{2}\mathbf{v}^T\mathbf{v} + \frac{1}{2}\left[\mathbf{H}\left(\mathbf{x}^b + \Delta\mathbf{x}\right) - \mathbf{y}^o\right]^T \mathbf{R}^{-1}\left[\mathbf{H}\left(\mathbf{x}^b + \Delta\mathbf{x}\right) - \mathbf{y}^o\right] + J_c(\mathbf{v}), \qquad (2)$$

There are six variables for the analysis vector $\mathbf{x}$, including the three wind components ($u$, $v$, and $w$), potential temperature ($\theta$), pressure ($p$), and water vapor mixing ratio ($q_v$). To assimilate reflectivity directly in the variational framework, hydrometeor-related model variables, including the mixing ratios for rainwater $q_r$, snow $q_s$, and hail $q_h$, are added to the analysis vector.

### 2.3.2. Dual-Resolution Hybrid 3DEnVAR Method

In the hybrid 3DEnVAR method, the augmentation of state vector $\mathbf{w}$ was added to the preconditioned incremental 3DVAR cost function in (2), yielding to:

$$J = \frac{1}{2}\mathbf{v}^T\mathbf{v} + \frac{1}{2}\mathbf{w}^T\mathbf{w} + \frac{1}{2}\left[H\left(\overline{\mathbf{x}}^b + \Delta\mathbf{x}\right) - \mathbf{y}^o\right]^T \mathbf{R}^{-1}\left[H\left(\overline{\mathbf{x}}^b + \Delta\mathbf{x}\right) - \mathbf{y}^o\right] + J_c, \qquad (3)$$

where:

$$\Delta\mathbf{x} = \Delta\mathbf{x}_1 + \Delta\mathbf{x}_2 = (\beta_1\mathbf{B})^{\frac{1}{2}}\mathbf{v} + (\beta_2\mathbf{P})^{\frac{1}{2}}\mathbf{w}, \qquad (4)$$

$\Delta\mathbf{x}$ is the analysis increment of the state of vector $\mathbf{x}$, $\mathbf{B}$ is the static 3DVAR background error covariance matrix, and $\mathbf{P}$ is the covariance matrix derived from the ensemble. To conserve the total background error, two positive coefficients $\beta_1$ and $\beta_2$ were used to determine the relative weights for the static background error covariance and the ensemble error covariance, assuming:

$$\beta_1 + \beta_2 = 1, \qquad (5)$$

This approach to combining two covariance matrices to form a hybrid covariance provides flexibility since it allows for different relative contributions from the two covariance matrices. When $\beta_1 = 1$, only the static 3DVAR background error covariance matrix is used for the assimilation (i.e., pure 3DVAR method), and, conversely, when $\beta_2 = 1$, a pure ensemble-derived covariance matrix is used.

Smaller ensemble sizes can lead to underdispersion and filter divergence. A larger ensemble helps improve the background error covariance estimation, but at a higher computational cost. To reduce the computational cost and create a larger ensemble, lower-resolution ensemble forecasts were used to estimate the background error covariance of

higher-resolution deterministic forecasts. This dual-resolution method is developed and implemented well in real-time analyses and forecast systems [53,65].

## 3. Experimental Design and Model Description

### 3.1. Experimental Design

A severe convective event associated with the Meiyu front on 30 June 2018, which occurred in the middle and lower reaches of the Yangtze River, was selected to examine the performance of the lightning and radar data assimilation (LRDA). The description of this severe convective event can be found in Liu et al. [35]. All the experiments examined herein, including a control experiment and multiple assimilation experiments, are listed in Table 1. As its name indicates, the control experiment (labeled CTL) does not assimilate any data. All assimilation experiments were performed using either single or multiple analysis cycles. A set of single-analysis experiments help provide a clearer assessment of the effects of 3DVAR and hybrid 3DEnVAR methods when assimilating lightning and/or radar data in the analysis field. The single-analysis experiments were performed at 0000 UTC on 30 June. The high-frequency cycling analysis experiments were designed to evaluate the ability of the combined assimilation of radar and lightning data to depict the evolution and movement of the convective-scale event of interest. The cycling assimilation experiments consist of successive analyses from 0000 UTC to 0100 UTC on 30June in 15 min intervals. In the single and cycling analysis experiments described above, the hybrid 3DEnVAR method use $\beta1 = 0.4$, $\beta2 = 0.6$, which were determine based on trial and error. To test the effect of different percentages of the static covariance matrix and ensemble covariance matrix in the hybrid method on the assimilation results, two additional assimilation experiments with different coefficients (cf., Table 1) for the hybrid 3DEnVAR method of assimilating the combined lightning and radar data were performed and only the forecast results were evaluated.

**Table 1.** Abbreviations used for the experiments and descriptions of the experiments. All assimilation experiments were performed using single and cycling analysis, respectively. Single-analysis experiments were at 0000 UTC on 30 June, and cycling analysis experiments were performed from 0000 to 0100 UTC on 30 June with 15 min frequency. In single-analysis experiments, lightning frequency was accumulated from 2300 UTC on 29 June to 0030 UTC on 30 June, and in cycling analysis experiments, lightning frequency was accumulated for the 15 minutes before the analysis moment. To test the different hybrid coefficients, two sets of hybrid coefficients were used to assimilate the combined lightning and radar data. The label "cov06" represents $\beta_1 = 0.4$ and $\beta_2 = 0.6$, label "cov08" represents $\beta_1 = 0.2$ and $\beta_2 = 0.8$, and the label "cov10" represents $\beta_1 = 0.0$ and $\beta_2 = 1.0$.

| Experiments | Data Assimilated | Data Assimilation Methods |
|---|---|---|
| CTL | None | None |
| LDA_3DVAR | FY-4A LMI | 3DVAR method ($\beta_1 = 1.0$, $\beta_2 = 0.0$) |
| LDA_Hybrid_cov06 | | Hybrid 3DEnVAR method, ($\beta_1 = 0.4$, $\beta_2 = 0.6$) |
| RDA_3DVAR | Radar reflectivity and radial velocity | 3DVAR method ($\beta_1 = 1.0$, $\beta_2 = 0.0$) |
| RDA_Hybrid_cov06 | | Hybrid 3DEnVAR method, ($\beta_1 = 0.4$, $\beta_2 = 0.6$) |
| LRDA_3DVAR | FY-4A LMI, radar reflectivity, and radial velocity | 3DVAR method ($\beta_1 = 1.0$, $\beta_2 = 0.0$) |
| LRDA_Hybrid_cov06 | | Hybrid 3DEnVAR method, ($\beta_1 = 0.4$, $\beta_2 = 0.6$) |
| LRDA_Hybrid_cov08 | | Hybrid 3DEnVAR method, ($\beta_1 = 0.2$, $\beta_2 = 0.8$) |
| LRDA_Hybrid_cov10 | | Hybrid 3DEnVAR method, ($\beta_1 = 0.0$, $\beta_2 = 1.0$) |

*3.2. Model Description*

The numerical model used for this study is the three-dimensional compressible non-hydrostatic Weather Research and Forecasting model version 3.7.1 (WRF3.7.1). The higher-resolution deterministic forecast uses a grid spacing of 3 km (Figure 1b) and 50 terrain-following eta levels from the surface extending to an assumed model top of 50 hPa. The higher-resolution deterministic forecast is derived from Global Forecast System (GFS) datasets with 6 h intervals and a 0.25° grid spacing. The main physics schemes used for the deterministic forecasts include the Thompson microphysical parameterization [69], the Dudhia scheme for shortwave radiation [70], and the Rapid Radiative Transfer Model scheme (RRTM) for longwave radiation [71], the Yonsei University planetary boundary layer scheme [72], and the unified Noah land surface model [73].

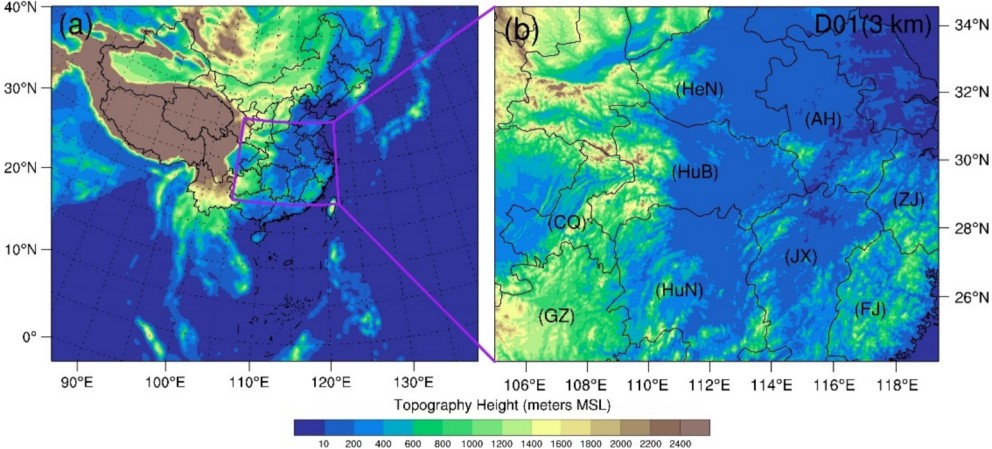

**Figure 1.** The little global map (**a**) and configuration of the WRF model domain with a grid spacing of 3 km (**b**). The colors represent model terrain heights. The abbreviations for the Henan, Anhui, Hubei, Chongqing, Guizhou, Hunan, Jiangxi, Zhejiang, and Fujian Provinces are HeN, AH, HuB, CQ, GZ, HuN, JX, ZJ, and FJ, respectively.

The lower-resolution ensemble members have a horizontal grid spacing of 9 km. The lower-resolution model domain encompasses the higher-resolution deterministic forecast model domain. The initial and boundary conditions of the lower-resolution ensemble members were provided by the 21 ensemble members from the National Centers for Environmental Prediction (NCEP) Global Ensemble Forecast System (GEFS) with 6 h intervals and a 1.0° grid spacing. To expand the ensemble sizes and improve spread, three groups of physical parameterization schemes are used to generate a total of 21 × 3 ensemble members. The different sets of microphysical, radiation, planetary boundary layer, and land surface model schemes are used for each group. To ensure that each ensemble includes convective scale information, the KF cumulus parameterization scheme was used for all members. All ensemble members were initialized at 1800 UTC on 29 June 2018 and integrated until 0600 UTC on 30 June. In this study, the 63 members of the ensemble did not assimilate any observations.

## 4. Results

In this severe convective event, there were two independent convective cells in the central part of Hubei Province at 0000 UTC on 30 June, which later merged into a mesoscale convection system at 0300 UTC. At 0000 UTC, a large amount of lightning occurred in the two convection cells [35]. The analysis results including radar reflectivity and the increments of wind, water vapor and hydrometer variables from the single-analysis experiments are shown and described in this section. The forecast results including radar reflectivity and accumulated precipitation from the single and cycling analysis experiments

were evaluated. The benefit of using the flow-dependent background error covariances on the analysis will be highlighted.

### 4.1. Analysis Field of Single-analysis Experiments

In the 3DVAR method, the analysis field is the state vector that minimizes the cost function (1). During the cost function minimization procedure, the iteration is stopped once the gradient of the cost function no longer decreases (i.e., converge to a minimum value). The total cost function is the sum of the cost of each individual term in (3). In this study, the single cost term includes the background field term, six variables ($u$, $v$, $w$, $\theta$, $p$, $q_v$), the divergence constraint term, and hydrometeor-related model variables to assimilate radar reflectivity. Generally, there are fewer conventional observations than radar observations. In the same domain, the conventional observation points are notably more sparsely distributed than the radar observation points that are interpolated into the model grid. This will cause the total cost function to always be dominated by the radar data term when other observations and radar data are assimilated simultaneously. Figure 2 shows the total cost function and summation of the gradient norm as a function of the number of iterations for all single-analysis experiments. When assimilating radar data, the total cost function is much larger than when only assimilating lightning-derived water vapor pseudo-observations. In the experiments of this study, to ensure that the minimum cost function is attained, 200 iterations are used. In Figure 2, it can be seen that after approximately 120 iterations, the curve of each experimental cost function converges well, and that the gradient drops notably (by 90% on average). In the experiment assimilating pseudo-observations for water vapor in the hybrid 3DEnVAR method, the gradient no longer drops after iteration 76, and the minimization procedure is halted automatically. In the next section, the detailed results of the analysis are presented, including radar reflectivity, wind field, water vapor, and hydrometers.

#### 4.1.1. Radar Reflectivity and Wind Field

Figure 3 shows the maximum reflectivity and horizontal wind at the single-analysis time (0000 UTC). The maximum reflectivity of the two convective cells exceeds 45 dBZ in the central and southeast regions of Hubei, and there is also a large area of strong reflectivity at the junction of the Anhui and Jiangxi provinces (Figure 3a). The control run fails to simulate the reflectivity in southeast Hubei and southeast Anhui (Figure 3b). In the LDA experiments, the reflectivity analyzed by the 3DVAR method is consistent with the background field (Figure 3c). In LDA_Hybrid_cov06, weaker reflectivity values are analyzed in southeastern Hubei. This underestimation of reflectivity by the 3DVAR relative to the observations is even more pronounced in southeastern Anhui, where the analysis increment of reflectivity barely exceeds 30 dBZ (Figure 3d). The analysis of reflectivity relies heavily on radar data because the assimilation of lightning data has very little effect on the hydrometeor variables (3DVAR no effect). Therefore, the reflectivity analysis fields of the RDA and LRDA experiments exhibit similar patterns and remain in reasonable agreement with the observations (Figure 3e–h). Additionally, the spurious reflectivity echoes present in the background in CTL over northern Hunan and western Hubei are suppressed (Figure 3e–h).

Whether the 3DVAR or hybrid 3DEnVAR assimilation methods are used, the reflectivity information of the analysis field can be well-analyzed when radar data are assimilated alone or combined with the lightning-derived pseudo-observations for water vapor. Concomitantly, because of the assimilation of radar radial wind information, some appropriate adjustments were made to the wind field in the analysis after assimilating the radar data. In severe convection forecasting, a better reflectivity analysis field will correspond to a shorter spin-up time. The combination of lightning and radar data can yield to a reflectivity analysis field more consistent with the observations and partially alleviate the problem of a poor reflectivity analysis when only assimilating lightning data. However, for areas lacking radar data coverage, obtaining better-analyzed reflectivity (hydrometers) relies on other

observation data, such as lightning data. In the experiment assimilating lightning-derived water vapor, improved reflectivity analyses are obtained through the hybrid 3DEnVAR method in southeastern Hubei and southeastern Anhui where radar observations are scarce (Figure 3d).

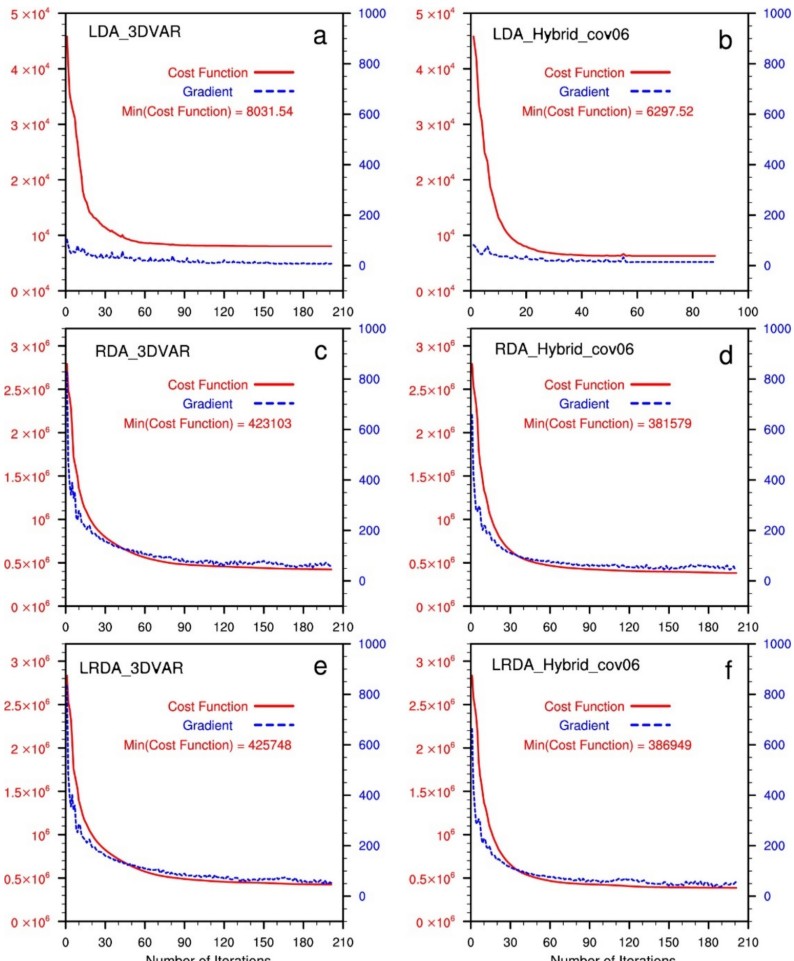

**Figure 2.** The total cost function and summation of gradient norm as a function of the number of iterations in single-analysis experiments. The red solid line is the cost function (left Y-axis), and the blue dashed line is the gradient norm (right Y-axis). (**a**,**b**) Lightning data assimilation by 3DVAR (LDA_3DVAR) and hybrid 3DEnVAR (LDA_Hybrid_cov06), (**c**,**d**) radar data assimilation by 3DVAR (RDA_3DVAR) and hybrid 3DEnVAR (RDA_Hybrid_cov06), and (**e**,**f**) the combined lightning and radar data assimilation by 3DVAR (LRDA_3DVAR) and hybrid 3DEnVAR (LRDA_Hybrid_cov06).

Figure 4 shows the horizontal and vertical cross-sections of increments of wind vectors and vertical velocity. The assimilation of the radar radial wind data provides an adjustment in the wind field. At the location where strong reflectivity values are observed on the border between the Anhui and Jiangxi provinces, convergence enhancement in the horizontal direction can be clearly noted (Figure 4a,c) along with well-defined vertical motions (Figure 4d,f). The wind field adjustments in the LRDA experiments are very similar to those in the RDA experiments (not shown). In particular, the adjustment of the wind field on the boundary between the Anhui and Jiangxi provinces is smaller when using the hybrid 3DEnVAR method relative to pure 3DVAR. When only assimilating the lightning-derived water vapor data using the pure 3DVAR method, the wind field could not be as accurately adjusted compared to the hybrid 3DEnVAR. In the LDA experiment using the hybrid 3DEnVAR method, the horizontal wind field in the analysis field was adjusted more accurately (Figure 4b,e). The adjustment of the wind field is important to trigger the observed convection earlier and to maintain. Wind field adjustments, however, will

occasionally trigger spurious cells, and overcoming this problem requires the assimilation of more information about the wind and a more detailed analysis of the wind field.

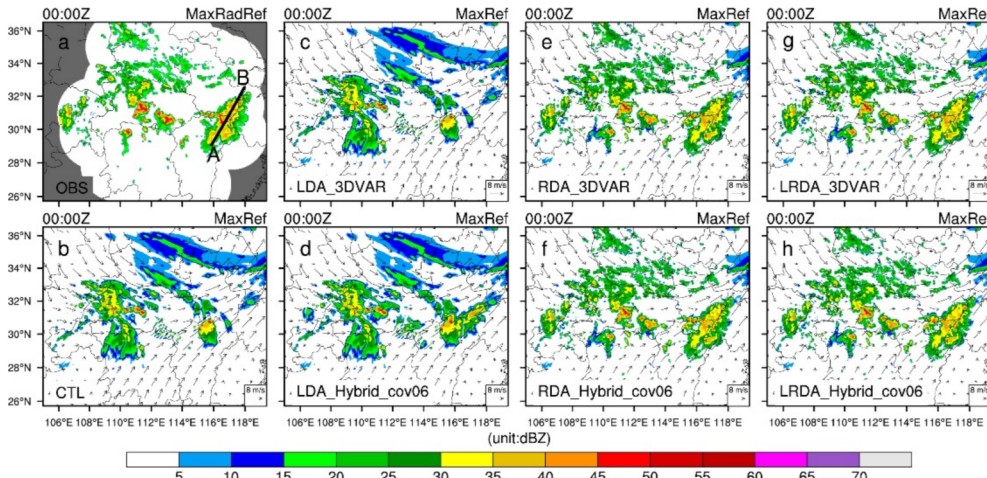

**Figure 3.** The observed maximum radar reflectivity (MaxRadRef) and analyzed maximum reflectivity (MaxRef) horizontal wind vectors at z = 4 km at 0000 UTC on 30 June 2018 (analysis time). (**a**) Observed maximum radar reflectivity interpolated onto the 3 km simulation domain, (**b**) control run (CTL), and for single-analysis experiments: (**c**,**d**) lightning data assimilation by 3DVAR and hybrid 3DEnVAR, (**e**,**f**) radar data assimilation by 3DVAR and hybrid 3DEnVAR, (**g**,**h**) and the combined lightning and radar data assimilation by 3DVAR and hybrid 3DEnVAR. The white background area is the range of radar scanning in (**a**). The black line AB in (**a**) denotes the locations of the vertical cross-sections for subsequent figures.

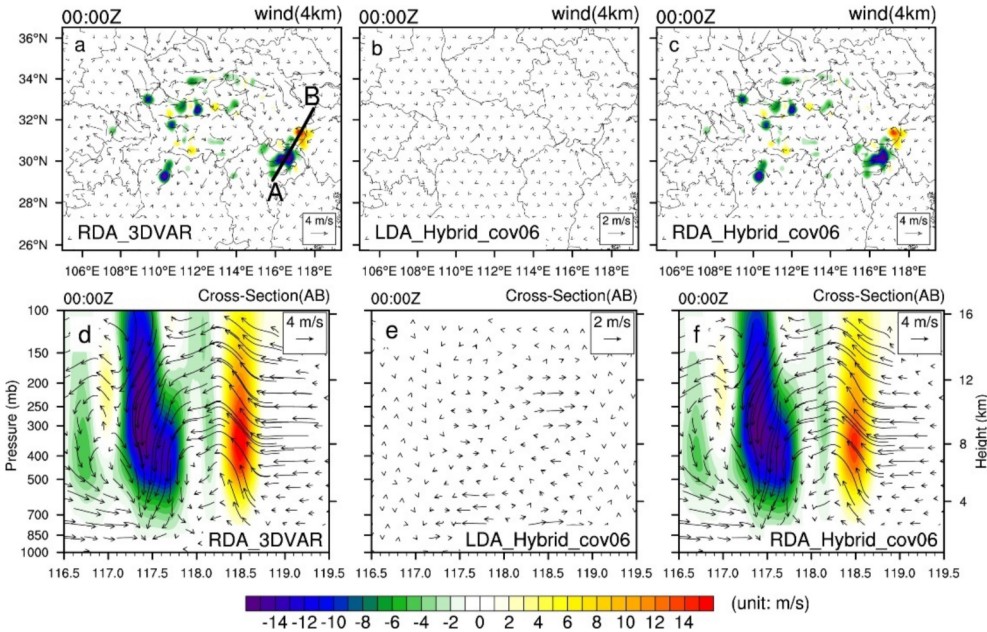

**Figure 4.** Horizontal increments of vertical velocity (shaded contours) and wind vector (black vector arrows) from single-analysis experiments at z = 4 km (**a**–**c**) and vertical-cross sections of increments of vertical velocity and wind vector (**e**–**f**) at 0000 UTC 30 June 2018 (analysis time) along line AB in (**a**). The line AB in (**a**) is the same position as the line AB in Figure 3a. Lightning data assimilation by hybrid 3DEnVAR (**b**,**e**), radar data assimilation by 3DVAR (**a**,**d**) and hybrid 3DEnVAR (**c**,**f**). The black lines AB in (**a**) denote the locations of the vertical cross-sections for (**d**–**f**).

#### 4.1.2. Water Vapor and Hydrometers

The analysis increments of $q_v$ between eta level 0–15 (15th eta level is approximately 5.7 km in height) were summed and shown in Figure 5a–f. In the LDA experiments, a positive $q_v$ adjustment is produced at the location where lightning occurred with the hybrid 3DEnVAR method producing the largest $q_v$ increase (Figure 5d), followed by the 3DVAR method (Figure 5a). For the same number of iterations, the hybrid 3DEnVAR method yields a smaller total cost function value than 3DVAR when assimilating only the lightning-derived water vapor and, therefore, generates the largest $q_v$ increment. The $q_v$ increment cannot be obtained when assimilating the radar data directly using the 3DVAR method (Figure 5b), but a small adjustment of $q_v$ occurs in the analysis field when assimilating radar data through the hybrid 3DEnVAR method (Figure 5e). Because of the relatively larger number of data points contained in volumetric radar scans relative to two-dimensional flash density fields, the cost function of the radar data is much larger than that of the lightning pseudo-water vapor, and, thus, dominates the total cost function. In the LRDA experiments, the $q_v$ increment is smaller than in the LDA experiments.

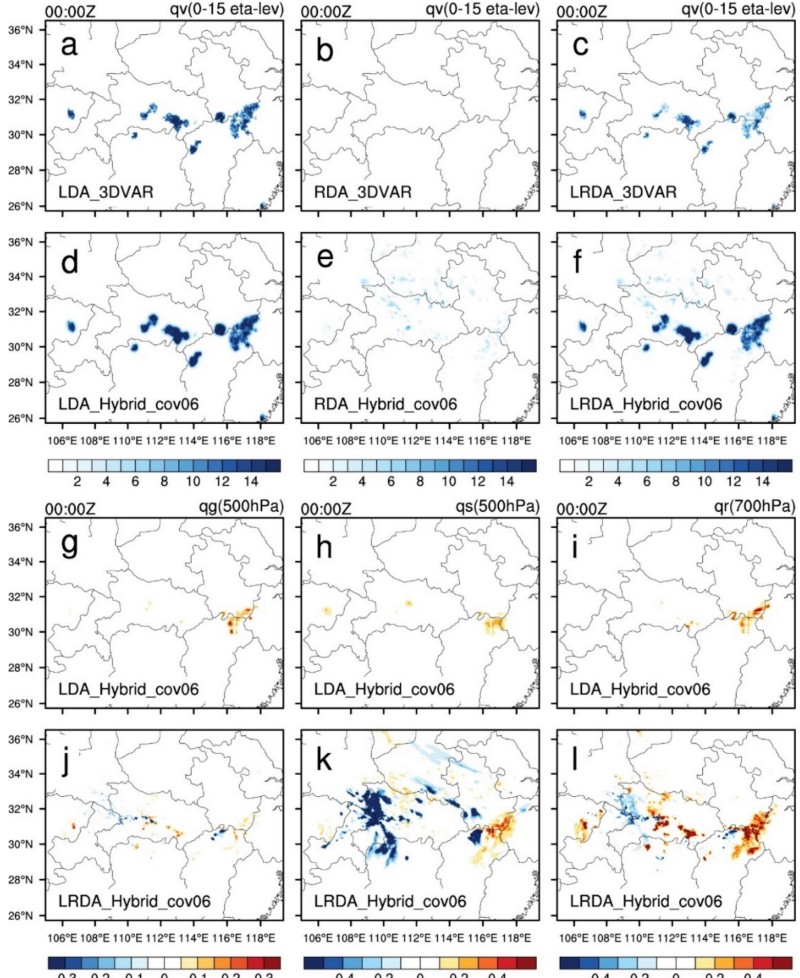

**Figure 5.** The single-analysis increments of water vapor mixing ratio ($q_v$) from 0 eta level to 15 eta level (**a**–**f**) were summed; graupel mixing ratio ($q_g$) at 500 hPa (**g**,**j**), snow mixing ratio ($q_s$) at 500 hPa (**h**,**k**), and rain mixing ratio ($q_r$) at 700 hPa (**i**,**l**) at 0000 UTC 30 June 2018 (analysis time). (**a**) Lightning data assimilation by 3DVAR. (**d**,**g**–**i**) Lightning data assimilation by hybrid 3DEnVAR. (**b**,**e**) Radar data assimilation by 3DVAR and hybrid 3DEnVAR. (**c**) The combined lightning and radar data assimilation by 3DVAR, and (**f**,**j**–**l**) uses hybrid 3DEnVAR.

Lightning data can provide water vapor information for convective development. However, there is large uncertainty in the empirical pseudo-observations for water vapor derived from the lightning data. To prevent larger errors from pseudo-observations being introduced into the analysis field, it is always desired that a minimum number of unreliable pseudo-observations are used to derive the analysis. When there is only a small increase in $q_v$, it is necessary to rely on hydrometeor information to trigger convection. When radar data are assimilated, the analysis consists of positive adjustments in the hydrometeor fields (Figure 5j–l). The increase in hydrometeor mass allows for an earlier development of convection in the analysis and, hence, a reduction in the spin-up time. The hybrid 3DEnVAR method analysis exhibits a small increment in hydrometeor mass over the northern portion of the Jiangxi province, while the 3DVAR method does not produce any hydrometeor increments (Figure 5g–i). After assimilating radar reflectivity data, a large negative increment of hydrometeors in western Hubei is noted, which indicates that the radar data (zero reflectivity) is able to partially suppress some of the spurious convection in the background field.

To further illustrate the changes in water vapor and hydrometeors, the vertical cross-sections of the analysis increments of $q_v$ and hydrometeors are shown (Figure 6). In the LDA_3DVAR and LRDA_3DVAR experiments, the $q_v$ increments maintain a consistent change in the vertical direction, with larger increments occurring at 600 and 850 hPa in the LDA_Hybrid_cov06 and LRDA_Hybrid_cov06 experiments. After assimilating radar reflectivity, positive $q_s$ increments of about 0.1 g/kg are produced above 500 hPa. At about 700 hPa, positive graupel mixing ratio ($q_g$) and $q_r$ increments of 0.1 and 0.3–0.6 g/kg are also seen, respectively.

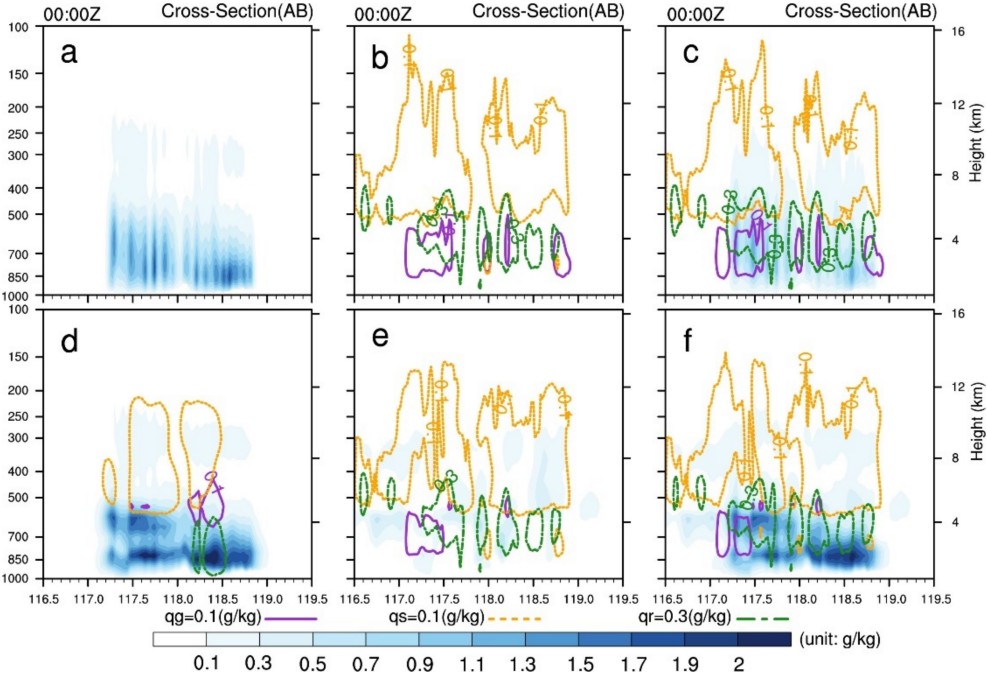

**Figure 6.** Vertical cross-sections of analysis increments of $q_v$ (blue shaded contour lines), $q_g$ (dark orchid contour lines), $q_s$ (orange contour lines) and $q_r$ (forest green contour lines) from single-analysis experiments at 0000 UTC 30 June 2018 (analysis time) along line AB in Figure 3a. (**a**,**d**) Lightning data assimilation by 3DVAR and hybrid 3DEnVAR, (**b**,**e**) radar data assimilation by 3DVAR and hybrid 3DEnVAR, and (**c**,**f**) the combined lightning and radar data assimilation by 3DVAR and hybrid 3DEnVAR.

*4.2. Forecast Field*

In this section, the forecasts of reflectivity and accumulated precipitation from the single and the multiple (cycling) analysis experiments were evaluated. To provide a more

complete view of the precipitation forecast performance, the hourly precipitation product combined with automatic weather stations precipitation observations in China and the CMORPH (Climate Prediction Center morphing method) satellite inversion precipitation product provided by the Chinese National Meteorological Information Center (NMIC) was used to calculate the precipitation forecast skill scores, including the equitable threat score (ETS) and categorical performance diagrams [74–76]. ETS = 1 indicates a perfect forecast while ETS = 0 stands for no forecast skill. The performance diagrams conveniently combine the information from the frequency bias, the probability of detection (POD), the critical success index (CSI), and the success ratio (one minus the false alarm rate). All score metrics were computed for neighborhood radii of 9 km (i.e., three grid points). The three thresholds (1, 5 and 10 mm/h) were used to indicate the occurrence of precipitation, heavy precipitation, and severe heavy precipitation, respectively.

### 4.2.1. The Single-analysis Experiments

Although CTL was arguably able to forecast the observed stronger reflectivity echoes in the central part of the Hubei Province in the 3 h forecast, some deviation from the observations can be seen in terms of position and intensity; these are located further to the northwest and are overall weaker than observed. A reflectivity exceeding 30 dBZ was observed in the western part of the Zhejiang province, but CTL failed to resolve it (Figure 7b). All DA experiments show a positive impact on the forecasting of composite radar reflectivity (Figure 7c–j). In the LDA experiments, the observed high-reflectivity areas in central Hubei were reasonably well-forecasted, but spurious cells are still present in the western portion of the Hubei Province (Figure 7c–f). Due to the lack of adjustment of the $q_v$ fields in RDA, the updated hydrometeor in the forecast field were shorter-lived. The improvements in the RDA experiment 3 h forecast compared with CTL were not obvious (Figure 7d–g). The LRDA experiments produced better forecast performances for reflectivity. The spurious cells seen in the LDA experiments in western Zhejiang are weaker in the LRDA experiments (Figure 7e,h–j), which also successfully forecasted the high-reflectivity region in central Hubei. In LRDA, the hybrid 3DEnVAR method reduced the spurious cells in western Hubei (Figure 7h–j). The reduction in spurious cells is more obvious in LRDA_Hybrid_cov06 compared to LRDA_Hybird_cov08 or LRDA_Hybrid_cov10.

Figure 8 shows horizontal cross-sections of observed and forecasted 6 h accumulated precipitation from the single-analysis experiments. From 0000 to 0600 UTC, there was a heavy rainfall band located in the central and southeastern of Hubei with accumulated precipitation exceeding 50 mm/6 h (Figure 8a). CTL was able to simulate a high-precipitation center in western Hubei but could not forecast the precipitation in southeastern Hubei (Figure 8b). After assimilating lightning, the observed precipitation band was better resolved but with a slight displacement and magnitude bias (Figure 8c,f). After assimilating radar data, the $q_v$ increment was small (Figure 5b,e), and the improvement in the 6 h accumulated precipitation was not obvious (Figure 8d,g). The radar data mainly adjusted the wind and hydrometeor fields, and there were obvious improvements in the forecasts for the first 1 to 2 h (not shown). In the LDA experiments, a stronger precipitation center than observed is seen at the junction of the Anhui and Zhejiang Provinces. In LRDA experiments, the increments in $q_v$ were overall smaller than that of LDA, which explains the smaller accumulated rainfall produced there.

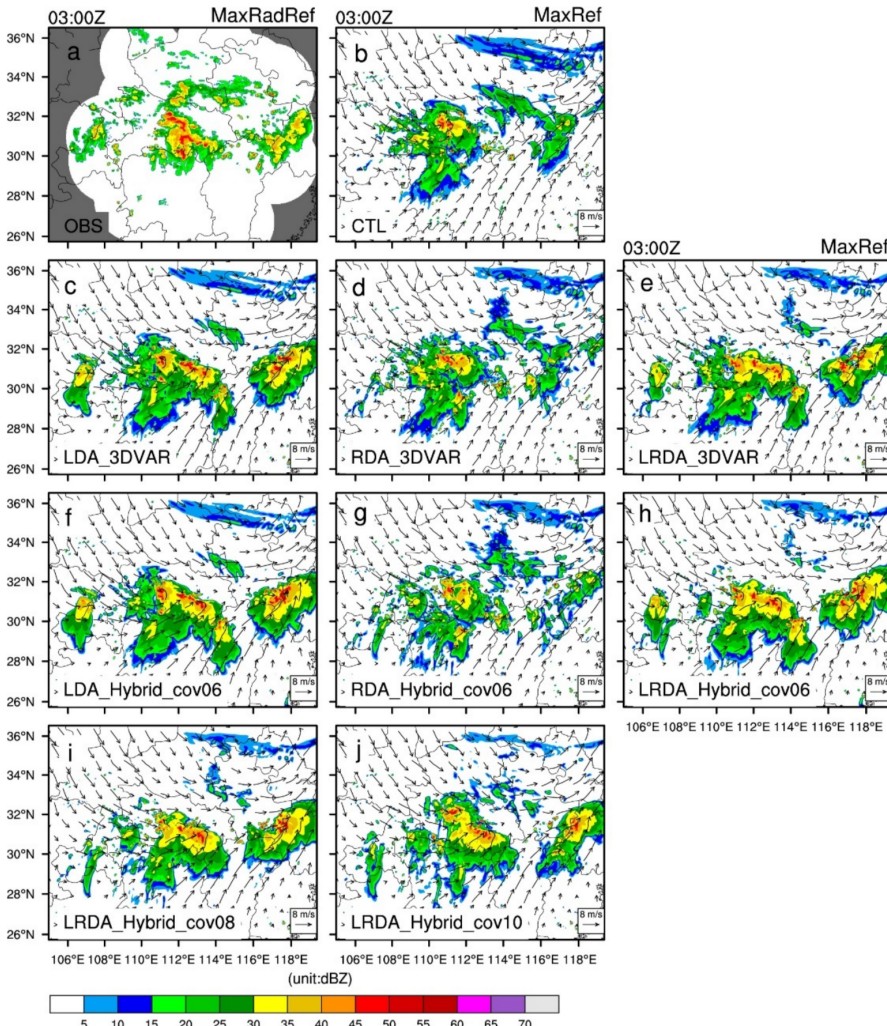

**Figure 7.** The observed maximum radar reflectivity and forecasted maximum reflectivity and horizontal wind vectors from single-analysis experiments at 0300 UTC on 30 June 2018 (i.e., 3 h forecast). (**a**) Observed maximum radar reflectivity interpolated onto the 3 km simulation domain (OBS), (**b**) control run (CTL), lightning (**c,f**), radar (**d,g**), and combined lightning and radar data assimilation (**e,h**) by 3DVAR (**c–e**) and hybrid 3DEnVAR (**f–h**) showed that $\beta_1 = 0.4$, $\beta_2 = 0.6$. The combined lightning and radar data assimilation by hybrid 3DEnVAR (**i,j**) showed that $\beta_1 = 0.2$, $\beta_2 = 0.8$ and $\beta_1 = 0.0$, $\beta_2 = 1.0$, respectively.

The precipitation skill score reflects more intuitively the precipitation forecast performances of different experiments. The ETSs (Figure 9a1–c3) of hourly accumulation precipitation indicated that either alone or in combination, the assimilation of lightning and/or radar data had an overall positive contribution to precipitation forecasting. In the first hour of forecast, the control run had ETS scores less than 0.2 at all thresholds, and all DA experiments had ETSs scores above or near 0.4. Compared with LDA and RDA, the combined lightning and radar data assimilation showed a more obvious improvement in the 6 h accumulated precipitation forecasts. At the 1 mm threshold, the ETS scores of LDA were less than 0.4. The ETS scores of RDA and LRDA were greater than or close to 0.6. Thanks to the radar-induced adjustments to the hydrometeor fields, the precipitation scores in the first hour of the LRDA experiment were higher than those of the LDA. In comparison to all DA experiments, the skill score for the hybrid 3DEnVAR method was higher than that of the 3DVAR method, except for the LRDA_Hybrid_cov10. The skill score for the first hours in the LRDA_Hybrid_cov10 experiment was worse than those produce by pure 3DVAR. This suggests that using a purely ensemble-derived covariance

statistics does not necessarily guarantee a superior forecast skill relative to forecasts using static background error covariance information. The performance diagrams for the 1 and 3 h forecast hourly accumulated precipitation are shown in Figure 9d1–e3. At the 1 mm threshold, the probability of detection is near 0.6 and the success ratio is less than 0.8 in LDA experiments. In RDA and LRDA experiments, the probability of detection is near 0.8 and the success ratio is greater than 0.8. In all experiments in the 1 h forecast, the LRDA_Hybrid_cov06 produced the highest CSI and POD for accumulated precipitation at 1 and 5 mm, together with a larger success ratio (Figure 9d1–d2). The forecast skill gradually decreases with increasing forecast time, and the LRDA_Hybrid_cov06 at 3 h forecast still produce good skill at 1mm, but the forecast performance at higher thresholds (i.e., 5 and 10 mm) is low (Figure 9e1–e3).

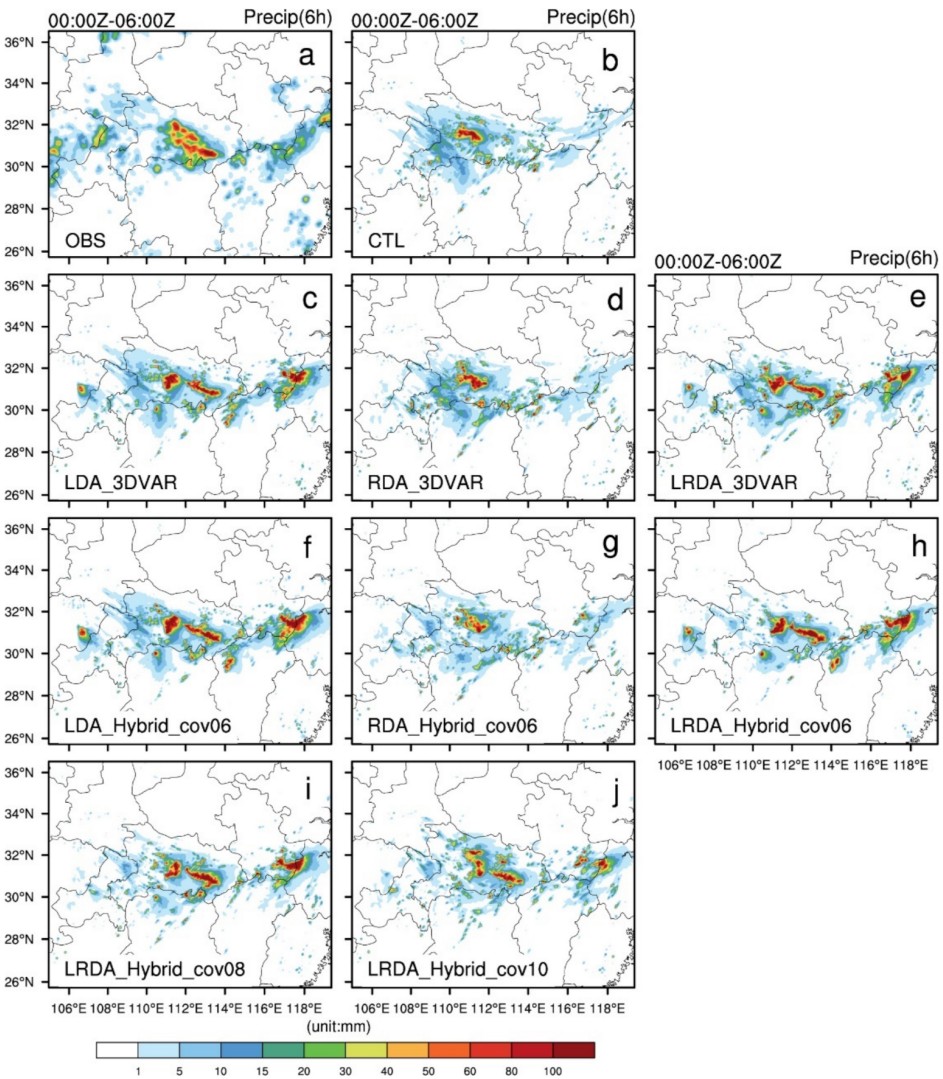

**Figure 8.** The observed and forecasted 6 h accumulated precipitation for single-analysis experiments from 0000 UTC to 0600 UTC on 30 June 2018. (**a**) Observed precipitation (OBS), (**b**) control run (CTL), lightning (**c**,**f**), radar (**d**,**g**) and the combined lightning and radar data assimilation (**e**,**h**) by 3DVAR (**c–e**) and hybrid 3DEnVAR (**f–h**) showed that $\beta_1 = 0.4$, $\beta_2 = 0.6$. The combined lightning and radar data assimilation by hybrid 3DEnVAR (**i**,**j**) showed that $\beta_1 = 0.2$, $\beta_2 = 0.8$ and $\beta_1 = 0.0$, $\beta_2 = 1.0$, respectively.

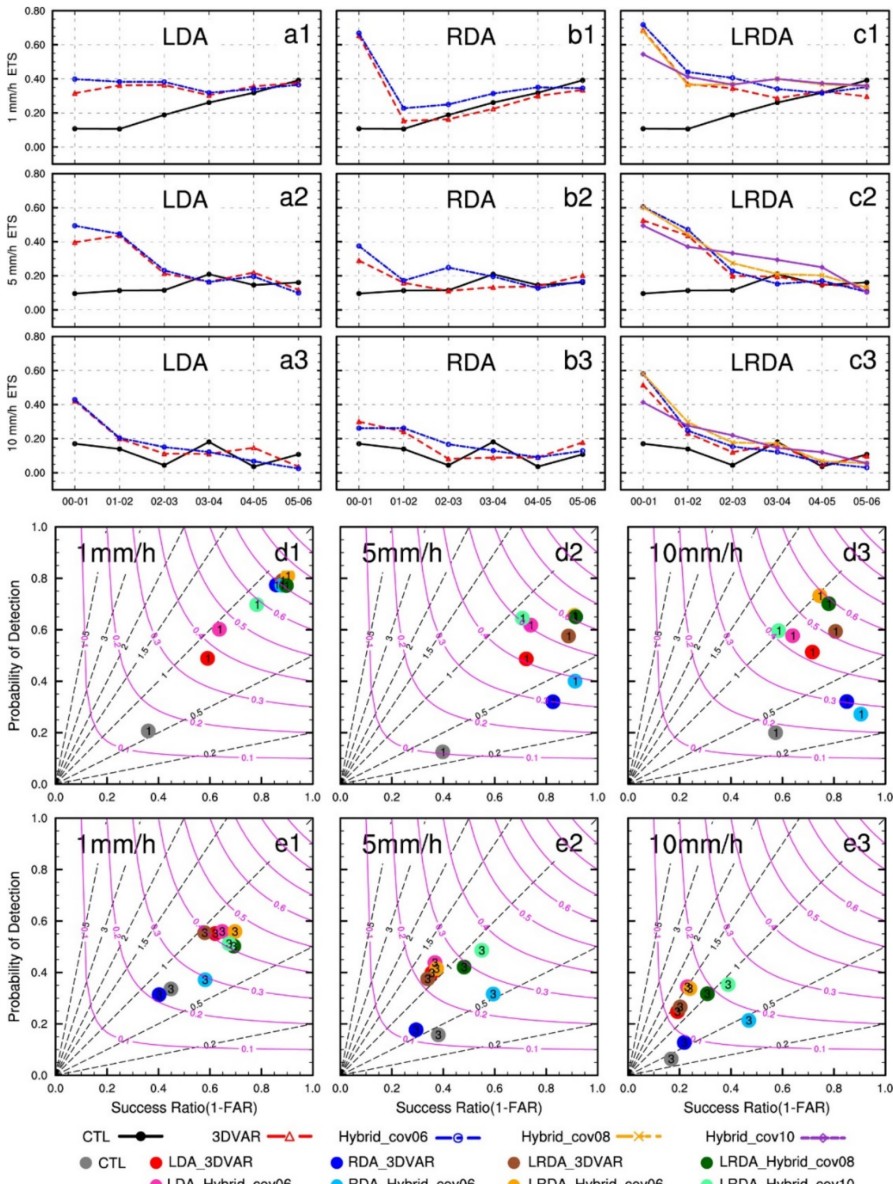

**Figure 9.** The equitable threat score (ETS) (**a1–c3**) of the forecasted hourly accumulated precipitation for single-analysis experiments from 0000 UTC to 0600 UTC on 30 June 2018. The performance diagram (**d1–d3,e1–e3**) of 1 and 3 h forecast hourly accumulated precipitation for single-analysis experiments from 0000 UTC to 0100 UTC and 0200 UTC to 0300 UTC on 30 June 2018. (**a1–a3**) The lightning data assimilation experiments (LDA), (**b1–b3**) the radar data assimilation experiments (RDA), and (**c1–c3**) the combined lightning and radar data assimilation experiments. (**a1,b1,c1,d1,e1**) the 1 mm threshold, (**a2,b2,c2,d2,e2**) the 5 mm threshold, and (**a3,b3,c3,d3,e3**) the 10 mm threshold. In each performance diagram plot, the lower-left corner represents no forecast skill and, similarly, the upper-right corner indicates perfect skill. Purple curves represent the critical success index (CSI), and the black dashed lines represent the frequency bias. The colored dots show the results for the experiments with legends shown at the bottom of the figure, the number inside each dot represents the forecast time in hours.

### 4.2.2. The Cycling Analysis Experiments

In the single-analysis experiments, one hour's worth of accumulated lightning data centered at the analysis time were assimilated in the LDA experiments, which may have a time representation error due to the relatively large accumulation time of 1 h instead of

10–15 min used for lightning. In the cycling analysis experiments, lightning data accumulated 15-min prior to the analysis time, were assimilated during each cycle. In other words, the 1 h lightning frequency information was ingested in the 3DVAR into four separate 15-min cycles. Figure 10 shows the maximum radar reflectivity for the 3 h forecast in the cycling analysis experiments. Similar to the results for the single-analysis experiments, the RDA exhibit overall smaller improvements in the reflectivity fields, with the LDA and the LRDA producing an overall positive impact on the reflectivity fields over Hubei.

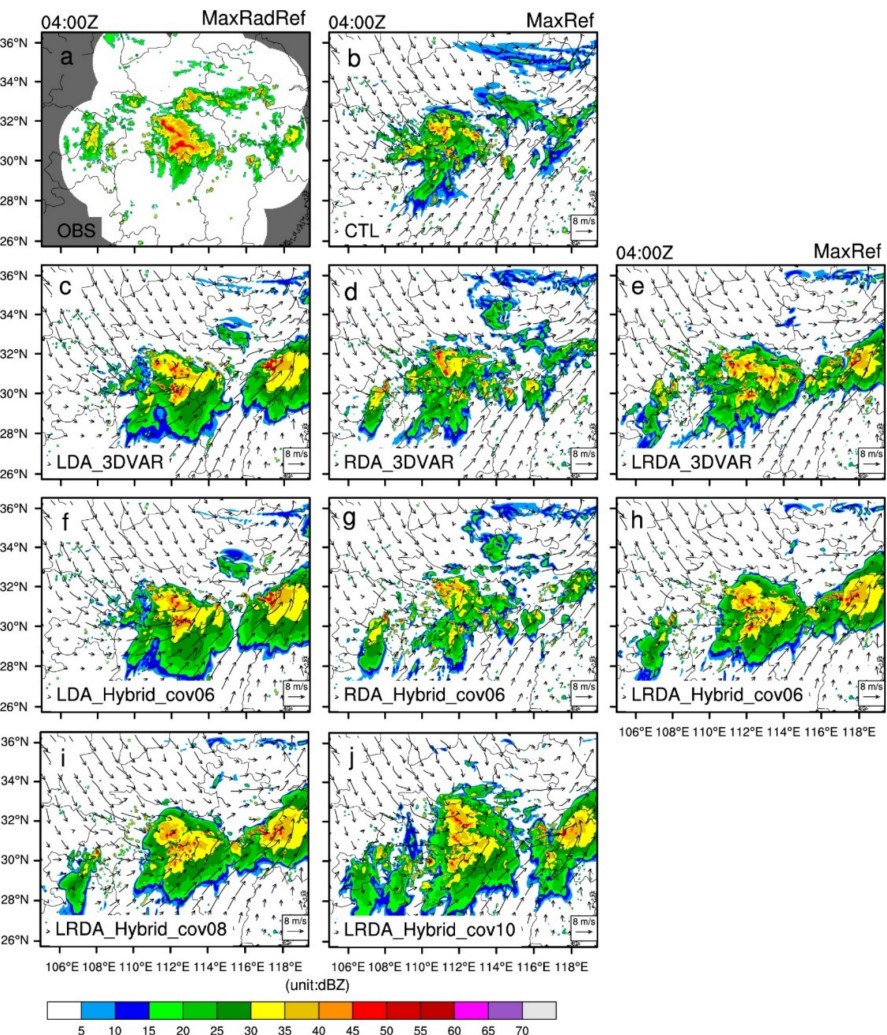

**Figure 10.** As in Figure 7, but for the cycling analysis experiments at 0400 UTC on 30 June 2018 (i.e., 3 h forecast). (**a**) Observed maximum radar reflectivity interpolated onto the 3 km simulation domain (OBS), (**b**) control run (CTL), lightning (**c**,**f**), radar (**d**,**g**) and the combined lightning and radar data assimilation (**e**,**h**) by 3DVAR (**c**–**e**) and hybrid 3DEnVAR (**f**–**h**) showed that $\beta_1 = 0.4$, $\beta_2 = 0.6$. The combined lightning and radar data assimilation by hybrid 3DEnVAR (**i**,**j**) showed that $\beta_1 = 0.2$, $\beta_2 = 0.8$ and $\beta_1 = 0.0$, $\beta_2 = 1.0$, respectively.

When examining the forecast results for 6 h accumulated precipitation, the range and magnitude of precipitation produced by the LDA are larger than those of the other assimilation tests (Figure 11), which is also consistent with the single-analysis experiments. Compared to the LRDA, the LDA generates a larger increment of water vapor with each cycling analysis, which amplifies this difference after multiple cycles and therefore yields to a more pronounced wet bias in the forecast. Similar precipitation forecast results were seen between LRDA_Hybrid_cov06 and LRDA_Hybrid_cov08, but the differences in LRDA_Hybrid_cov10 are more obvious. There are many unobserved heavy precipi-

tation points in the LRDA_Hybrid_cov10 forecast (Figure 11j) scattered throughout the domain. Overall, the hourly rainfall forecast skill of the cycling analysis experiments was superior to those using a single-analysis (Figure 12). Four hours before the forecast, the ETSs score in the LDA and LRDA cycling analysis experiments occasionally reached or exceeded 0.3 at 1 mm (Figure 12a1,c1) and 5mm (Figure 12a2,c2). In the 1mm threshold performance diagram, the LRDA produced the highest success ratio and POD in the 1 h forecast (Figure 12d1). In the 3 forecast, both LDA and LRDA produced higher POD than RDA (Figure 12e1). The combination of both pseudo-observations for water vapor derived from the lightning and the volumetric radar information is crucial to improve precipitation forecast in the early hours. The precipitation forecast in the following hours is found to be dependent on the pseudo-observations for water vapor.

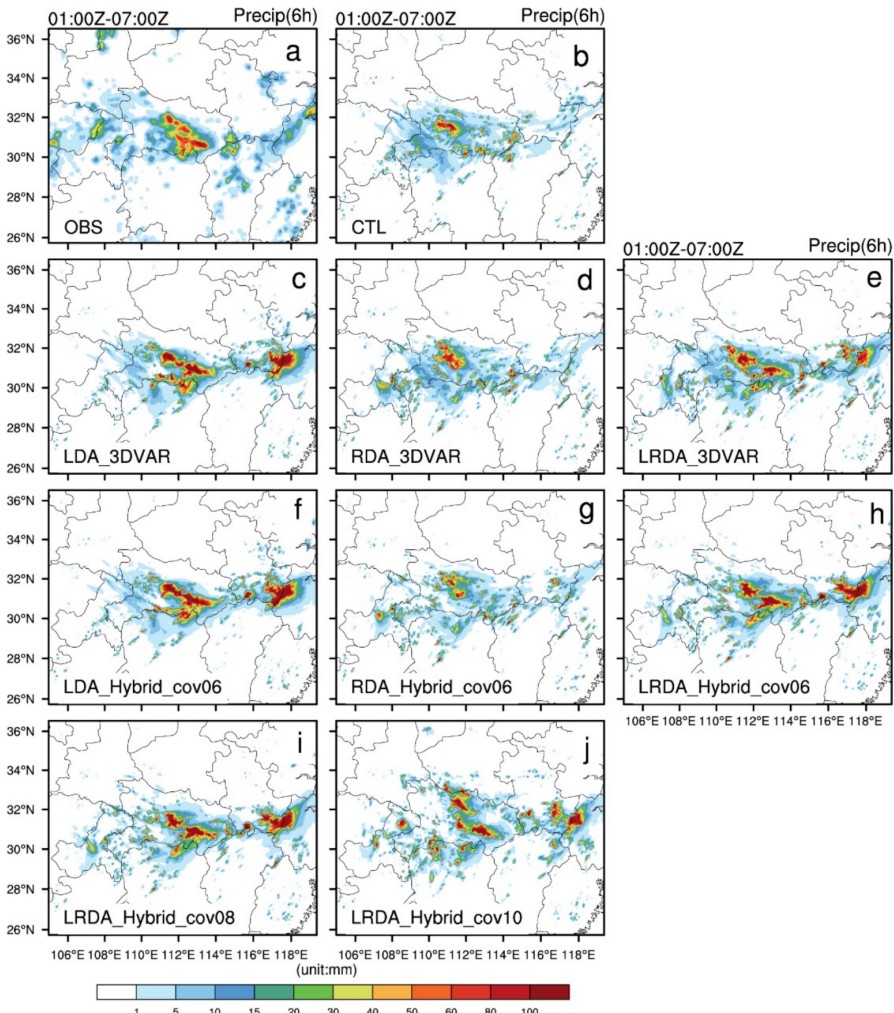

**Figure 11.** As in Figure 8, but for the cycling analysis experiments from 0100 UTC to 0700 UTC on 30 June 2018. (**a**) Observed precipitation (OBS), (**b**) control run (CTL), (**b**) control run (CTL), lightning (**c,f**), radar (**d,g**) and the combined lightning and radar data assimilation (**e,h**) by 3DVAR (**c–e**) and hybrid 3DEnVAR (**f–h**) showed that $\beta_1 = 0.4$, $\beta_2 = 0.6$. The combined lightning and radar data assimilation by hybrid 3DEnVAR (**i,j**) showed that $\beta_1 = 0.2$, $\beta_2 = 0.8$ and $\beta_1 = 0.0$, $\beta_2 = 1.0$, respectively.

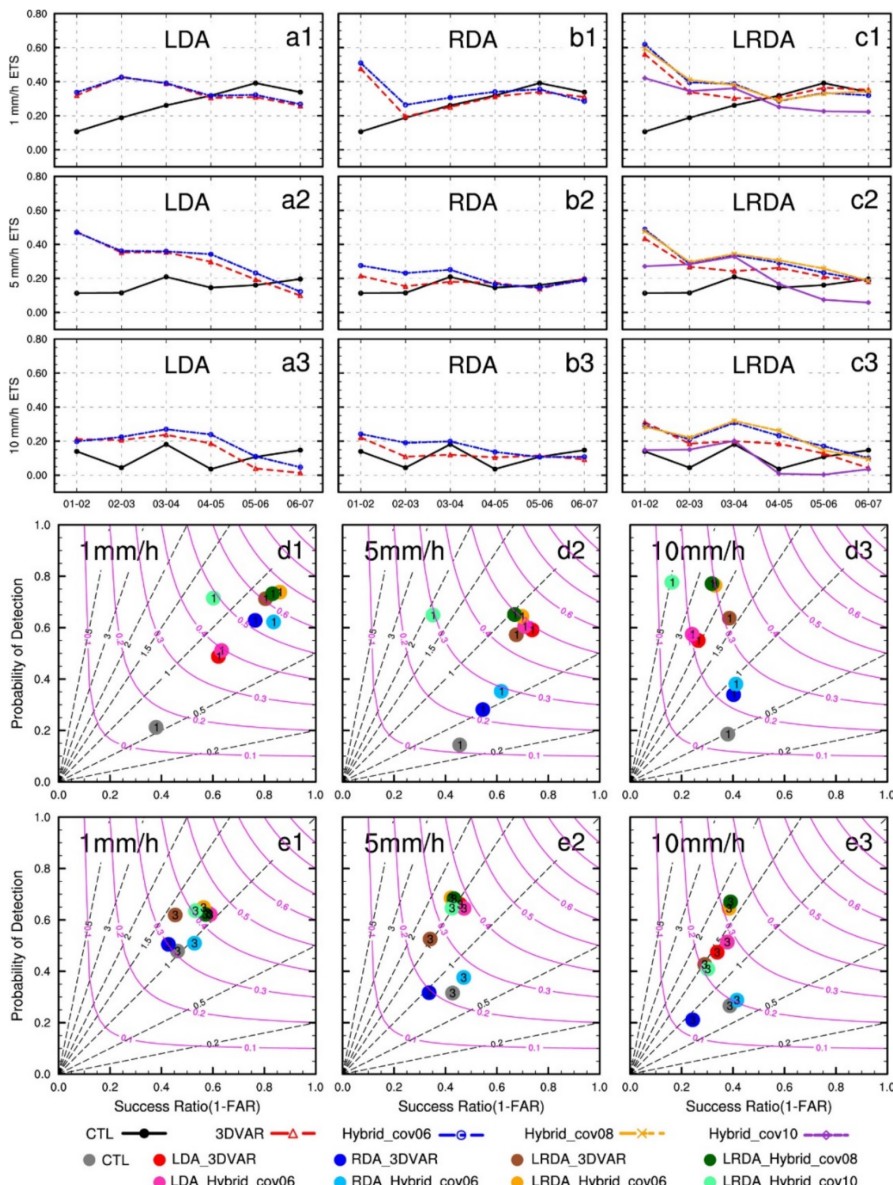

**Figure 12.** As in Figure 9, but for the cycling analysis experiments. (**a1**–**a3**) the lightning data assimilation experiments (LDA), (**b1**–**b3**) the radar data assimilation experiments (RDA), (**c1**–**c3**) the combined lightning and radar data assimilation experiments. (**a1**,**b1**,**c1**,**d1**,**e1**) the 1 mm threshold, (**a2**,**b2**,**c2**,**d2**,**e2**) the 5 mm threshold, (**a3**,**b3**,**c3**,**d3**,**e3**) the 10 mm threshold.

## 5. Summary and Conclusions

In this work, the performance of combined radar and lightning data assimilation in a dual-resolution hybrid 3DEnVAR system was examined through a set single and cycling analysis experiments. The sensitivity of the coefficients of covariance matrix was tested in assimilation experiments of combined lightning and radar data by the hybrid 3DEnVAR method. A severe convective event associated with the Meiyu front in the middle and lower reaches of the Yangtze River was used to evaluate the forecast performance of the assimilation experiment.

The single-analysis results showed that in the LDA experiments, the increase in $q_v$ was the largest. The RDA experiments showed large hydrometeor and vertical velocity increases but almost no $q_v$ increase. In the LRDA experiments, there was a moderate increase in $q_v$ coupled with large hydrometeor and vertical velocity increases. Since radar reflectivity provides information about the distribution of various kinds of hydrometeors, water vapor

increments cannot be obtained when directly assimilating radar reflectivity using the 3DVAR method. The pseudo-water vapor created by the lightning data complements the lack of water vapor information in the direct assimilation of radar reflectivity well. In addition, when assimilating the lightning and radar data simultaneously, the radar information dominates the total cost function, so the water vapor increment obtained by LRDA is smaller than that of LDA. In the LDA experiments, the maximum radar reflectivity in the analysis field was almost the same as that in the control experiment. The maximum reflectivity in the analysis was consistent with the observations in the RDA and LRDA experiments.

Overall, the results of the cycling analysis experiments are superior to those using a single-analysis. The precipitation skill score for LDA in the first forecast hour was low, and there was spurious convection developing at 3 h forecast that led to errors in the 6 h accumulated precipitation forecast. In the RDA experiments, the precipitation scores in the first hour of forecast were relatively high, but the improvements in the forecasts of the 3 h and 6 h accumulated precipitation were not obvious. In the LRDA experiments, the precipitation skill score at 1 h forecast was higher than LDA. Additionally, the 3 h forecast for reflectivity and 6 h accumulated precipitation remained consistent with the observations. Additionally, several areas characterized by spurious reflectivity and precipitation were partially removed. Whether the simulation assimilated lightning or radar data alone or both, the hybrid 3DEnVAR method showed notable benefits over the 3DVAR method. In the sensitivity test focusing on the weight given to the ensemble vs static background error covariances, LRDA_Hybrid_cov06 produced the best forecast skill throughout the simulation while LRDA_Hybrid_cov08 was overall better at resolving heavy precipitation in the cycling analysis experiments.

When radar data were added to the LDA, the hydrometeor information provided by the radar reflectivity helped to trigger convection earlier, which reduced the amount of water vapor added onto the background and helped reduce the spin-up time. The radar radial velocity also provided some beneficial contributions to the dynamic field of the environment (especially in the vertical). Dual-resolution hybrid 3DEnVAR methods are able to leverage ensemble-based background error covariance statistics with a more affordable computational cost. This advantage was highlighted in the increase in $q_v$ and hydrometeors in the analysis field, which had more appropriate magnitudes, and in the dynamic fields of the environment. In the cycling experiments, more observations were introduced into the analysis field by successive smaller adjustments with observation data closer in time to that of the analysis. This approach may be more advantageous than introducing a larger number of observations in one single-analysis.

This work aimed to address several shortcomings involved in the usage of water vapor as a proxy variable in convective-scale lightning data assimilations. First, the lightning data from the LMI of the stationary satellite determined the position of lightning-active, deep mixed-phase convection in the plane view, but when deriving the pseudo-observation for water vapor, the adjustment range in the vertical direction was determined by an empirical height/depth. Second, when convection was activated by the local increases in water vapor mass in the background field, if the local increase remained small, longer spin-up time was required; if the amount of water vapor added was too large, wet biases could be exacerbated and could also lead to large areas of spurious storms. Third, in the LDA, the lack of adjustment to the environmental fields caused the convective system that was activated in the analysis field to often be short-lived. The first problem was addressed in Liu et al.'s (2020) work, and the same approach was applied to this work. The FY-4A cloud top height was used as the upper limit of water vapor adjustment to obtain more accurate pseudo-observations suitable for different convective modes. The use of the dual-resolution hybrid 3DEnVAR method with multivariant flow-dependent background error covariances combined with radar data in the lightning assimilation procedure can alleviate the second and third limitations well.

In this work, a hybrid ensemble convective-scale DA method was used to assimilate observations from multiple sources, which had positive impacts on the analysis. However, the lower-resolution ensemble members did not assimilate any observations and relied only on the cumulus parameterization scheme to obtain convective-scale information. Future work should focus on examining if improvements in the quality of the lower-resolution ensemble via the assimilation of lightning and/or radar data could help improve the skill of the higher-resolution deterministic forecasts further. In addition, if the lower-resolution ensemble assimilates the same observations with the same assimilation methods, could this lead to under dispersion in the ensembles? This issue will be further studied.

**Author Contributions:** Conceptualization, P.L. and Y.Y.; investigation, P.L.; data curation, P.L.; methodology, P.L. and Y.Y.; project administration, Y.Y.; writing—original draft, P.L.; writing—review and editing, P.L., Y.Y., A.L., Y.W., A.O.F., J.G. and C.W. All authors have read and agreed to the published version of the manuscript.

**Funding:** This work was supported by the National Key Research and Development Program of China (No. 2017YFC1502102) and the National Natural Science Foundation of China (No. 41175092) and NOAA/Office of Oceanic and Atmospheric Research under NOAA-University of Oklahoma Cooperative Agreement NA11OAR4320072.

**Institutional Review Board Statement:** Not applicable.

**Informed Consent Statement:** Not applicable.

**Acknowledgments:** This work supported by the Supercomputing Center of Lanzhou University. The authors thank the Chinese National Satellite Meteorological Center (NSMC) for providing the FY-4A lightning data and FY-4A cloud top height data (http://satellite.nsmc.org.cn/PortalSite/Data/Satellite.aspx, 30 June 2018) and the Chinese National Meteorological Information Center (NMIC) for providing the precipitation product (http://data.cma.cn/data/cdcdetail/dataCode/SEVP_CLI_CHN_MERGE_CMP_PRE_HOUR_GRID_0.10.html, 30 June 2018). The NCEP GFS analysis data are available from the NCAR Research Data Archive (RDA, https://rda.ucar.edu/datasets/ds084.1/index.html#!description, 30 June 2018). The National Oceanic and Atmospheric Administration (NOAA) for providing the GEFS dataset (https://www.ncdc.noaa.gov/data-access/model-data/model-datasets/global-forcast-system-gfs, 30 June 2018).

**Conflicts of Interest:** The authors declare no conflict of interest.

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
