# Peer review of "Assimilating FY-4A Lightning and Radar Data for Improving Short-Term Forecasts of a High-Impact Convective Event with a Dual-Resolution Hybrid 3DEnVAR Method"

_remotesensing, doi:10.3390/rs13163090_

Round 1
Reviewer 1 Report
General Comments to the authors:
In this study, the authors aim to evaluate the skill of the WRF model to predict the precipitation events exploring the data assimilation of lightning and radar data in a hybrid 3DEnVAR method. To do this, the comparison with the 3DVAR method is done in a single and cycle experiments. The analysis and predict fields are evaluated and ETS statistical values in the different thresholds of precipitation intensity are explored to evaluate the different experiments of data assimilation and configuration of hybrid data assimilation. The current version of the manuscript s very well wright, however, there are some simple changes in the presentation of results that would improve the final version of the manuscript. In my opinion, this manuscript after minor improvement can be published in the Remote sensing Journal. See my comments in the "Specific Comments to the authors" section for more details about these points.
Specific Comments to the authors:
There are few scientific points with minor improvements that deserve the
attention of the authors in the revision of this paper to improve the final version of the manuscript:
1- In the abstract: the authors do not specify which dual-resolution is used
in this study (3 and 9 km, as can see in the lines 253 and 266,
respectively), please include this information in this part of the
manuscript.
2- The introduction section: is very complete with 76 papers mentioned.
However, at many points, these references are not suitable explored,
which the specific contribution of each work is presented to readers. I
think that the each reference would be better explored. Some good
examples of this problem are the references in lines 54, 47, and 106 of
the manuscript. In line 66, Papadopoulos et al. and Mansell et al. are the
references 19-21 and not 19-23, as is written in line 69. Please, improve
this revision.
3- Table 1: as suggest the equation 5, β1 plus β2 is equal a 1, but in three
parts of this table the sum of these values is not equal a 1, for example,
β1=1 and β2=0.6. This is correct in the LRDA_Hybrid_Cov10, which
β1=0 and β=1. Please, correct this inconsistency in this table.
4- In several part of the manuscript Hybrid data assimilation strategy
receive the label “cov” preceded of the value of β2*10. The first time of
this is presented in the manuscript is in Table 1, but nothing is sed about
this label. Please include a comment about this in the label of table 1,
and justifying the choose the β2 in this label.
5- Figure 1: please, put the little global plot and highlights the region where
the experiments were done.
6- Figure 3: the results obtained by radar data assimilation (3e and 3f) are
very impressive and show that the radar data are more significant in this
study. A natural question is: why the combined radar and lightning data
assimilation (LRDA experiments) are not presented in this figure? Please
include two plots with this information.
7- Figure 4: Please, include line AB in the superior plots of this figure (the
same in figure 3).
8- Figure 7, I suggest change the position of LRDA_Hybrid_cov08 from the
top to the left side of LRDA_hybrid_cov10, to make easier the
comparison of results presents by Hybrid strategy using different
coefficients.
9- Figure 8, figure 10, Figure 11: the same change suggested in Figure 7.
Reviewer 2 Report
To the authors:
I appreciate the effort by the authors to clearly present and explain their results, such that the paper has only minimal need for revision.
Still, of concern, though, is the organization of the summary and conclusions section. This section should begin with "In this work.." line 583, with additional result related details provided within the text that remains to the end of the paragraph at line 590 (e.g., the text location withing an below line 598).
The top part of this paragraph (beginning at 569 should follow, where the method is briefly described. I would suggest as a format to list the different methods tried, the main result and their significance, and then recommendations for future development or use.
Minor corrections:
178: The radar data were quality controlled which procedure includes buddy checking, --> controlled, whose procedure... Please define "buddy checking."
221-222: This dual-resolution method is found and implemented well in real-time analyses and forecast systems [47, 59]. --> Found and implemented well -- hard to follow this word sequence.
226:
under the influence of the Meiyu front on 30 June 2018 was selected to... --> Please rephrase.
245: Please spell the word "minutes" correctly.
283: At 0000 UTC, more frequent lightning occurred in the two convective cells [32].--> than when? To what are the authors comparing to?
336: Please mention what "CR" means in the figure legend.
374: Is it really necessary to show a,c, e and g, since no shading is shown?
377: Lighting --> Lightning
380: The horizontal cross sections of the sum of analysis increments --> The analysis increments were summed and shown in the horizontal...
395: Same as above.
425: It is unclear what the colored lines show (not the blue contouring).
469: "lighting --> lightning.
489: Figure legend might be rewritten: Same as Fig. 7, but for...
515: Extra space in the word hourly.
555: In the 1mm threshold performance diagram, the LRDA produced high success ratio and POD at 1-h forecast (Figure 12d1), 556 while at 3-h forecast both LDA and LRDA produced high POD (Figure 12e1) --> These sentences are not phrased correctly. The high success ratio and POD: maybe: At 1-h forecast time... "a high success ratio and high POD... Start a new sentence for phrase: the forecast at 3-h.
570: an attempt was made to propose solutions based on the data and methods. --> Not sure what the authors mean by this statement. Aren't the solutions part of the methods? What do you mean by the word methods in this context?
575: Second, it was difficult to determine the total amount of qv mass ingested into the analysis field during LDA. --> Why would this be difficult? Just sum the additional mass within the routine...?
591-92: The single analysis results showed that in the LDA experiments, the increase in qv was the largest. The RDA experiments showed large hydrometeor and vertical velocity increases but almost no qv increase. --> Please discuss in greater detail the significance of these results. Also "qv was the largest" of what? The fact that qv had almost no increase: does this mean that qv was converted into other particles, or just wasn't added?
597: The assimilation experiments have a similar result for precipitation forecasting in both single and cycling analysis approach. --> This is a poorly worded sentence as well as being vague (non-descriptive).
599: The precipitation skill score for LDA in the first forecast hour... --> I would suggest a new paragraph here and to provide more specific information (e.g., numbers -- what is "low?).
Reviewer 3 Report
Revision of the article titled:” Assimilating FY4 Lightning and Radar Data for Improving 2 Short-Term Forecasts of a High-Impact Convective Event with a 3 Dual-Resolution Hybrid 3DEnVAR Method “
The article presents an study on assimilation method by using 3DenVAR and combining static and ensemble background error covariances to assimilate radar data, and pseudo-water vapor observations to improve short-term severe weather forecasts with the Weather Research and Forecast (WRF) model.
In this study, the water vapor is estimates by combining total lightning data and cloud top height from a Fengyum Geostationary satellite.
I found the article very well written, the ideas are well thought and organized. I have some specific comments that I raise below.
I consider the artiche is aceptable for publication with minor corrections
Specific comments and suggestions:
1.- Lightning has been used no just to forecast precipitation, it has been umployed to estimate convective precipitation as well, I suggest the authors add a paragrapgh mentioning this fact in the introduction here are some recent studies about this:
- Xu, R. F. Adler, and N.-Y. Wang, “Improving geostationary satellite rainfall estimates using lightning observations: Under- lying lightning-rainfall-cloud relationships,” Journal of Applied Meteorology and Climatology, vol. 52, no. 1, pp. 213–229, 2013.
In some different way, these authors use A Kalman Filter to estimate convective precipitation
- Minjarez-Sosa, C.M.; Castro, C.L.; Cummins, K.L.; Waissman, J.; Adams, D.K. An improved QPE over complex terrain employing cloud-to-ground lightning occurrences. Appl. Meteorol. Climatol. 2017, 56, 2489–2507 

- Minjarez-Sosa C, Waissman J, Castro CL, Adams D. Algorithm for Improved QPE over Complex Terrain Using Cloud-to-Ground Lightning Occurrences. Atmosphere. 2019; 10(2):85. https://doi.org/10.3390/atmos10020085
2.- Line 82 the authors wrote: “Observational and laboratory studies have long shown an unambiguous association between ice-phase particles/graupel content and lightning” ……add reference
3.- At line 283 please change convection by convective
4.- In Figure 2 I suggest the authors to add what is the horizontal axis even if they think that is redundant
5.- Line 323 In order to not repeat the Word “large”, Perhaps the authors can write phrase in other way.
6.- Figure 2: Although the authors show a map with the name of the regions in figure 1 it could be better for the readers of this paper to show in these figures the regions of Hubei, Anhui and Hunan.
7.- Figure 9: I am wondering why the authors do not separate this figure in two, because the cost? I feel that is too much information posted in one figure, later in the text it can be explained and can be provided an analysis relating the two figures.
8.- Figure 9: line 515. Is something missing here??
10.- Lines- 618-620: How long takes the process? I am interested on the computational cost
